# The Impact of Initialization on LoRA Finetuning Dynamics

**Soufiane Hayou**
Simons Institute
UC Berkeley
hayou@berkeley.edu

**Nikhil Ghosh**
Dept of Statistics
UC Berkeley
nikhil_ghosh@berkeley.edu

**Bin Yu**
Dept of Statistics
UC Berkeley
binyu@berkeley.edu

## Abstract

In this paper, we study the role of initialization in Low Rank Adaptation (LoRA) as originally introduced in Hu et al. [19]. Essentially, to start from the pretrained model as initialization for finetuning, one can either initialize $B$ to zero and $A$ to random (default initialization in PEFT package), or vice-versa. In both cases, the product $BA$ is equal to zero at initialization, which makes finetuning *starts* from the pretrained model. These two initialization schemes are seemingly similar. They should in-principle yield the same performance and share the same optimal learning rate. We demonstrate that this is an *incorrect intuition* and that the first scheme (initializing $B$ to zero and $A$ to random) on average yields better performance compared to the other scheme. Our theoretical analysis shows that the reason behind this might be that the first initialization allows the use of larger learning rates (without causing output instability) compared to the second initialization, resulting in more efficient learning of the first scheme. We validate our results with extensive experiments on LLMs.

## 1 Introduction

The pretrain-finetune paradigm (e.g., [7, 9]) has revolutionized deep learning, replacing task-specific models trained from scratch with finetuning of pretrained base models. These base models, trained on generic unsupervised objectives, learn powerful features that can be rapidly adapted to downstream tasks. The most effective models are consistently the largest ones [14, 25], with state-of-the-art models reaching hundreds of billions of parameters. While many such models are openly available (e.g., Llama by Touvron et al. [38]), full finetuning remains computationally prohibitive for most practitioners. This has led to parameter-efficient finetuning methods, including adapters [11], prompt tuning [20], and $(IA)^3$ [24].

Low Rank Adaptation (LoRA) [19] has emerged as a leading parameter-efficient method, training only low-rank adapter matrices added to pretrained weights, typically using Adam [3]. LoRA often matches or exceeds full-finetuning performance [35, 39], though it may underperform on complex generation tasks. While prior work has examined rank [31] and learning rate [44] hyperparameters, initialization schemes remain understudied. This work provides experimental and theoretical justification for choosing between seemingly equivalent initialization approaches.

In standard LoRA training, one of the two LoRA matrices is initialized with random values and the other is initialized to zero (see Section 2.1). Recently, in Meng et al. [48] the authors proposed an alternative initialization scheme to LoRA which uses the top singular vectors of the pretrained weights as opposed to a random initialization and showed improved training on several tasks. To further improve LoRA training with quantization, Li et al. [34] introduced a new method called LoftQ for computing a better initialization for quantized training [27]. However, to the best of our knowledge,

38th Conference on Neural Information Processing Systems (NeurIPS 2024).

there has not been any study concerning the random initialization in vanilla LoRA. Specifically, it is not clear from prior work *which of the two LoRA matrices should be initialized to be zero*. Empirical results by Zhu et al. [50] suggested that the two initialization schemes mentioned above yield similar performance, but it is not clear if the learning rate was well-tuned for each initialization scheme. Our findings suggest that these two initialization schemes lead to fundamentally different finetuning dynamics, and that one of these schemes generally yields better result compared to the other.

**LoRA Variations.** Beyond altering the LoRA initialization scheme, there have been a series of works which try to address limitations of vanilla LoRA using different variations. To further reduce the number of trainable parameters, LoRA-FA [42] freezes the $A$ matrix which leads to small performance loss while reducing memory consumption by up to $1.4\times$. The performance of this training scheme is also investigated in Zhu et al. [50]. VeRA [33] freezes random weight tied adapters and learns vector scalings of the internal adapter activations. LoRA-XS [43] initializes the $A$ and $B$ matrices using the SVD of the pretrained weights and trains a low-rank update of the form $BRA$ where $R$ is a trainable $r \times r$ matrix and $B$, $A$ are fixed. NOLA [32] parametrizes the adapter matrices to be linear combinations of frozen random matrices and optimizes the linear coefficients of the mixtures. VB-LORA [46] shares adapter parameters using a global vector bank. In order to improve the learning ability for more challenging finetuning tasks, Kalajdzievski [31] proposes a scaling rule for the scalar adapter multiplier to unlock increased gains with higher adapter ranks. MoRA [45] learns high-rank updates while still preserving parameter efficiency by applying hand-designed compress and decompress operations before and after a trainable adapter matrix. DoRA [47] decomposes the pretrained weight into magnitude and direction components to allow for better training dynamics.

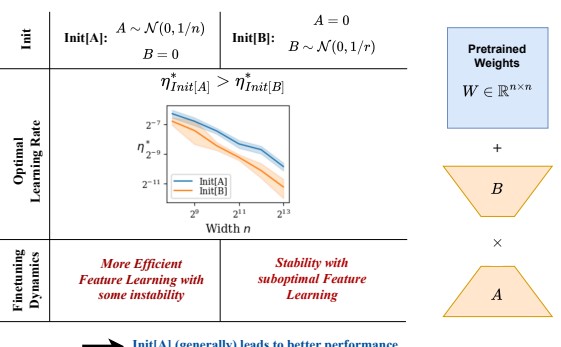

Figure 1: Summary of our contributions in this paper: a description of the difference between the finetuning dynamics when LoRA weights $A$ and $B$ are initialized with `Init[A]` or `Init[B]`.

**Contributions.** We study the impact of Initialization in LoRA through a theory of large width for neural networks. The core approach is to take the width of a neural network to infinity and determine how the behavior of the limit depends on the choice of the hyperparameters, such as the learning rate and initialization. This approach allows to derive principled scaling choices for these hyperparameters such that desired properties (e.g. stable feature learning) are achieved as the network size grows (see Appendix A.2 for more details). Examples of the infinite-width limit include works on initialization (e.g. He et al. [4]), and training dynamics (e.g. [21]). Examples for the depth limit include initialization strategies [6, 10, 30], and depth scaling (see e.g. [18, 23, 28, 29, 37, 41]). A similar strategy was used to derive scaling rules for the LoRA learning rate in Hayou et al. [44] (LoRA+) that concluded that the learning rates for different LoRA matrices should be scaled differently to ensure optimal feature learning. In this work we use the same approach to provide a systematic comparison between two different random initialization schemes for vanilla LoRA finetuning (using the same learning rate for the $A$ and $B$ matrices). Using the notation `Init[A]` to refer to the case where $A$ is initialized to random and $B$ to zero (as in [19]) and `Init[B]` for the opposite, we show that `Init[A]` and `Init[B]` lead to fundamentally different training dynamics (as shown in Figure 1):

1. `Init[A]` allows the use of larger learning rates compared to `Init[B]`

2. `Init[A]` leads to 'internal instability' where the features $Az$ (for some input $z$) are large but LoRA output $BAz$ is small. This form of instability allows more efficient feature learning. We identify a *feature learning / stability tradeoff* in this case.

3. `Init[B]` does not cause any instabilities but training is suboptimal ($B$ is undertrained).

4. Empirical results confirm the theory and show that `Init[A]` generally leads to better performance than `Init[B]`.

## 2 Setup and Definitions

We consider a general neural network model of the form

$$
\begin{cases}
Y_{in}(x) = W_{in}x, \\
Y_l(x) = \mathcal{F}_l(W_l, Y_{l-1}(x)), \ l \in [L], \\
Y_{out}(x) = W_{out}Y_L(x),
\end{cases}
\tag{1}
$$

where $x \in \mathbb{R}^d$ is the input, $L \geq 1$ is the network depth, $(\mathcal{F}_l)_{l \in [L]}$ are mappings that define the layers, and $W_l \in \mathbb{R}^{n \times n}$ are the hidden weights, where $n$ is the network *width*, and $W_{in}, W_{out}$ are input and output embedding weights.[1] This model will represent the pretrained model that will later be finetuned on some new task.

To finetune a (large) pretrained model with a limited amount of computational resources, a popular resource efficient approach is to use the LoRA finetuning method defined below.

**Definition 1** (Low Rank Adapters (LoRA) from [19]). *To apply LoRA to a weight matrix $W \in \mathbb{R}^{n_1 \times n_2}$ in the model, we constrain its update in the fine-tuning process by representing the latter with a low-rank decomposition $W = W^* + \frac{\alpha}{r}BA$. Here, only the weight matrices $B \in \mathbb{R}^{n_1 \times r}$, $A \in \mathbb{R}^{r \times n_2}$ are trainable and the original pretrained weights $W^*$ remain frozen. The rank $r \ll \min(n_1, n_2)$ and $\alpha \in \mathbb{R}$ are tunable constants.*

As the width $n$ grows,[2] the network initialization scheme and the learning rate should be adapted to avoid numerical instabilities and ensure efficient learning. For instance, the variance of the initialization weights (in hidden layers) should scale like $1/n$ to prevent the pre-activations from blowing up as we increase model width $n$ (e.g., He initialization [4]). To derive proper scaling rules, a principled approach consist of analyzing the statistical properties of key quantities in the model (e.g. second moment of the pre-activations) as $n$ grows and then adjust the initialization variance, the learning rate, and the architecture to achieve desirable properties in the limit $n \to \infty$ [5, 10, 13, 40]. We use this approach to study the effect of initialization on the feature learning dynamics of LoRA in the infinite-width limit. For more details about the theory of scaling of neural networks, see Appendix A.2.

Throughout the paper, we will be using asymptotic notation to describe the behaviour of several quantities as the width $n$ grows. Note that the width $n$ will be the only scaling dimension of neural network training which grows and all other scaling dimensions such as the LoRA rank $r$, number of layers $L$, sequence length, number of training steps, etc., will be considered as fixed. We use the following notation for the asymptotic analysis.

**Notation.** Given sequences $c_n \in \mathbb{R}$ and $d_n \in \mathbb{R}^+$, we write $c_n = \mathcal{O}(d_n)$, resp. $c_n = \Omega(d_n)$, to refer to $c_n < \kappa d_n$, resp. $c_n > \kappa d_n$, for some constant $\kappa > 0$. We write $c_n = \Theta(d_n)$ if both $c_n = \mathcal{O}(d_n)$ and $c_n = \Omega(d_n)$ are satisfied. For vector sequences $c_n = (c_n^i)_{1 \leq i \leq k} \in \mathbb{R}^k$ (for some $k > 0$), we write $c_n = \mathcal{O}(d_n)$ when $c_n^i = \mathcal{O}(d_n^i)$ for all $i \in [k]$, and same holds for other asymptotic notations. Finally, when the sequence $c_n$ is a vector of random variables, convergence is understood to be convergence in second moment ($L_2$ norm).

### 2.1 Initialization of LoRA Adapters

The standard way to initialize trainable weights is to take an iid initialization of the entries $A_{ij} \sim \mathcal{N}(0, \sigma_A^2)$, $B_{ij} \sim \mathcal{N}(0, \sigma_B^2)$ for some $\sigma_A, \sigma_B \geq 0$ (this includes initialization with zeros if $\sigma_B$ or $\sigma_A$

---

[1]We use the same notation from Hayou et al. [44].

[2]The width in SOTA models is typically large, i.e. of width $n > 10^3$.

are set to 0).[3] Due to the additive update structure of LoRA, we want to initialize the product $BA$ to be 0 so that finetuning starts from the pretrained model [19]. This can be achieved by initializing one of the weights $A$ and $B$ to 0. If both are initialized to 0, no learning occurs in this case since this is a saddle point and the parameter gradients will remain zero. Thus, we should initialize one of the parameters $A$ and $B$ to be non-zero and the other to be zero. If we choose a non-zero initialization for $A$, then following standard initialization schemes (e.g., He Init [4], LeCun Init [1]), one should set $\sigma_A^2 = \Theta(n^{-1})$ to ensure $Ax$ does not explode for large $n$. This is justified by the Central Limit Theorem (CLT). On the other hand, if we choose a non-zero initialization for $B$, one should make sure that $\sigma_b^2 = \Theta(r^{-1}) = \Theta(1)$. This leaves us with two possible initialization schemes:

- `Init[A]`: $\sigma_B^2 = 0, \sigma_A^2 = \Theta(n^{-1})$ (default initialization in LoRA [19]).
- `Init[B]`: $\sigma_B^2 = \Theta(r^{-1}) = \Theta(1), \sigma_A^2 = 0$.[4]

These two initialization achieve the goal of starting finetuning from the pretrained model. A priori, it is unclear if there is a material difference between the two initialization schemes. Surprisingly, as we will show later in this paper, these two initialization schemes lead to fundamentally *different training dynamics* when model width is large.

## 2.2 LoRA Features

**Notation.** For a given LoRA layer in the network, we use $\underline{Z}$ to denote the input to that layer and $\bar{Z}$ for the output after adding the pretrained weights. More precisely, we can write the layer operation as $\bar{Z} = W^* \underline{Z} + \frac{\alpha}{r} BA \underline{Z}$.

Our main analysis relies on a careful estimation of the magnitude of several quantities involving *LoRA features*. Let us first give a formal definition.

**Definition 2** (LoRA Features). *Given a general neural architecture and a LoRA layer (Definition 1), we define LoRA features $(Z_A, Z_B)$ as $Z_A = A\underline{Z}$, and $Z_B = BZ_A = BA\underline{Z}$. At fine-tuning step $t$, we use the superscript $t$ to denote the value of LoRA features $Z_A^t, Z_B^t$, and the subscript $t$ to denote the weights $A_t, B_t$.*

# 3 LoRA Finetuning Dynamics in the Large Width Limit

We fix the LoRA rank $r$ throughout the analysis and examine the finetuning dynamics in the limit of large width. This setup aligns well with practical scenarios where the rank is much smaller than the width (i.e., $r \ll n$). Typically, for Llama models the rank $r$ is generally of order $2^k$ for $k \in \{2, \ldots, 6\}$, and model width $n$ is generally larger than $2^{12}$. We will refer to a layer of the network to which LoRA is applied (see Definition 1) as a *LoRA layer*. For the theoretical analysis, we adopt a simplified setting that facilitates a rigorous yet intuitive derivations of the results.

## 3.1 Simplified Setting

The following simplified setup was considered in Hayou et al. [44] to derive asymptotic results concerning the learning rates in LoRA. We use the same setup in our analysis to investigate the impact of initialization.

**Finetuning Dataset.** We assume that the dataset used for finetuning consists of a single datapoint $(x, y)$,[5] and the goal is to minimize the loss calculated with the model with adjusted weights $W^* + BA$ for all LoRA layers (here $\theta = \{A, B, \text{for all LoRA layers in the model}\}$). $\underline{Z}^t$ is the input to the LoRA layer, computed with data input $x$. Similarly, we write $d\bar{Z}^t$ to denote the gradient of the loss function with respect to the layer output features $\bar{Z}$ evaluated at data point $(x, y)$.

---

[3]Gaussianity is not important and can be replaced by any zero-mean distribution with finite-variance for our purposes.

[4]Here, we assumed that $r = \Theta(1)$ (in width), i.e. it doesn't grow with width. In general, the right scaling for `Init[B]` is $\sigma_B^2 = \Theta(r^{-1})$.

[5]Although this a simplifying assumption for our analysis, the results can be extended to mini-batched gradients without affecting the conclusions. Such results will require additional assumptions to be fully rigorous.

**Single LoRA Module.** Given a LoRA layer, LoRA feature updates are not only driven by the change in the $A, B$ weights, but also the changes in $\underline{Z}, d\bar{Z}$ which are updated as we finetune the model (assuming there are multiple LoRA layers). To isolate the contribution of individual LoRA layers to feature learning, we assume that only a *single LoRA layer is trainable* and all other LoRA layers are frozen.[6] For this LoRA layer the layer input $\underline{Z}$ is fixed and does not change with $t$, whereas $d\bar{Z}$ changes with step $t$ (because $\bar{Z}^t = (W^* + \frac{\alpha}{r}B_t A_t)\underline{Z}$). After step $t$, $Z_B$ is updated as follows

$$\Delta Z_B^t = \underbrace{B_{t-1}\Delta Z_A^t}_{\delta_t^1} + \underbrace{\Delta B_t Z_A^{t-1}}_{\delta_t^2} + \underbrace{\Delta B_t \Delta Z_A^t}_{\delta_t^3}. \tag{2}$$

As discussed in Hayou et al. [44], the terms $\delta_t^1, \delta_t^2$ represent 'linear' feature updates that we obtain if we fix one weight matrix and only train the other. The third term $\delta_t^3$ represents the 'multiplicative' feature update which captures the compounded update due to updating both $A$ and $B$.

### 3.2 Stability and Feature Learning

Hayou et al. [44] introduced the notion of stability of LoRA features as width grows. We introduce here a slightly more relaxed notion of stability.

**Definition 3** (Feature Stability). *We say that LoRA finetuning is stable if for all LoRA layers in the model, and all training steps $t$, we have $\underline{Z}, Z_B = \mathcal{O}(1)$, as the width $n$ goes to infinity.*

Here, feature stability implies that LoRA output $Z_B$ remains bounded (in $L^2$ norm) as width grows. To achieve such stability, hyperparameters (initialization, learning rate) should be scaled as $n$ grows. We will show that the dependence of the optimal learning rate on $n$ is highly sensitive to the choice of initialization (`Init[A]` or `Init[B]`).

Note that feature stability also requires that $\underline{Z} = \mathcal{O}(1)$ which is directly related to pretraining dynamics since it depends on some pretrained weights $W^*$. We assume that pretraining parameterization (how initialization and learning rate are parametrized w.r.t width) ensures this kind of stability (see Appendix A for more details).[7]

As discussed above, feature updates are driven by the terms $(\delta_t^i)_{i \in \{1,2,3,\}}$. As $n$ grows, these feature updates might become trivial (i.e. vanish as $n \to \infty$) or unstable (i.e. grows unbounded). To avoid such scenarios, we want to ensure that $\Delta Z_B = \Theta(1)$. Such conditions are the main ideas behind $\mu$P [26] and Depth-$\mu$P [41], which are network parametrizations that ensure stability and feature learning in the large width and depth limits for pretraining. We recall this definition from [44].

**Definition 4** (Feature Learning). *We say that LoRA finetuning induces stable feature learning in the limit of large width if the dynamics are stable (Definition 3), and for all finetuning steps $t$, we have $\Delta Z_B^t \overset{def}{=} Z_B^{t+1} - Z_B^t = \Theta(1)$.*

$\Delta Z_B$ is the sum of the terms $\delta_t^i$'s (Equation (2)). To achieve optimal feature learning, we want to ensure that $\delta_t^1 = \Theta(1)$ and $\delta_t^2 = \Theta(1)$ which means that both weight matrices $A$ and $B$ are efficiently updated and contribute to the update in $Z_B$. An intuitive explanation is provided in Appendix A.1. This leads us to the following definition of efficient learning with LoRA.

**Definition 5** (Efficient Learning with LoRA). *We say that LoRA fine-tuning is efficient if it is stable (Definition 3), and for all LoRA layers in the model, and all fine-tuning steps $t > 1$, we have*

$$\delta_t^i = \Theta(1), \quad i \in \{1, 2\}.$$

Next, we introduce the $\gamma$-operator, an essential tool in our analysis of the large width dynamics.

---

[6]This is equivalent to having only a single LoRA layer in the model since LoRA layers are initialized to zero.

[7]When taking the infinite width limit, we can for instance assume that pretraining parameterization is $\mu$P [26]. This is a technicality for the infinite-width limit and does not have any implications on practical scenarios where the width is finite. The most important implications of this assumption is that in the pretrained network (before introducing LoRA layers), we have $\underline{Z} = \Theta(1), \bar{Z} = \Theta(1)$, which holds for a general input-output pair $(x, y)$.

### 3.3 Introduction to the $\gamma$-operator

In the theory of scaling, one usually tracks the asymptotic behavior of key quantities as we scale some model ingredient. For instance, if we scale the width $n$ of a neural network, we are interested in quantifying how certain quantities in the network behave as $n$ grows. This is a standard approach for (principled) model scaling and it has so far been used to derive scaling rules for initialization [5], activation function [10], network parametrization [41], amongst other things.

With Init[A] and Init[B], initialization weights are of order $\Theta(n^{-\beta})$ for some $\beta \geq 0$. Assuming that the learning rate also scales polynomialy with $n$, it is straightforward that preactivations, gradients, and weight updates are all asymptotically polynomial in $n$. Note that this is only possible because all neural computations consists of sums of $\Theta(n^{\alpha})$ terms, where typically $\alpha \in \{0, 1\}$. For instance, when calculating the features $A\underline{Z}$, each entry is a sum of $n$ terms, while when calculating $BZ_A$, each entry is a sum of $r$ terms ($r$ fixed as $n$ goes to infinity). This is true for general neural computation that can be expressed as Tensor Programs [15].

Consequently, for some quantity $v$ in the computation graph, it is natural to track the exponent that determines the asymptotic behavior of $v$ with respect to $n$. We write $v = \Theta(\gamma[v])$ to capture this polynomial dependence. Elementary operations with the $\gamma$-operator include:[8]

**Zero.** When $v = 0$, we write $\gamma[v] = -\infty$ (as a limit of $\gamma[n^{-\beta}]$ when $\beta \to \infty$).

**Multiplication.** Given two real-valued variables $v, v'$, we have $\gamma[v \times v'] = \gamma[v] + \gamma[v']$.

**Addition.** Given two real-valued variables $v, v'$, we *generally* have $\gamma[v + v'] = \max(\gamma[v], \gamma[v'])$. The only case where this is violated is when $v' = -v$. This is generally a zero probability event if $v$ and $v'$ are random variables that are not perfectly (negatively) correlated, which is the case in most situations where we make use of this formula. See Appendix A.3 for discussion.

We have now introduced all required notions for the subsequent analysis. For better readability, we defer all the proofs to the appendix.

### 3.4 Recursive formulas

Using the $\gamma$-operator, we can track the asymptotic behavior of the finetuning dynamics as model width $n$ grows. At finetuning step $t$, the weights are updated as follows

$$A_t = A_{t-1} - \eta g_A^{t-1}, \quad B_t = B_{t-1} - \eta g_B^{t-1},$$

where $g_A, g_B$ are processed gradients (e.g. normalized gradients with momentum as in AdamW). We assume that the gradients are processed in a way that makes their entries $\Theta(1)$. This is generally satisfied in practice (with Adam for instance) and has been considered in [40] to derive the $\mu$-parametrization for general gradient processing functions. From this, we obtain the following recursive formulas for $\gamma[Z_A^t]$ and $\gamma[B_t]$, which characterizes their behavior in the large width limit.

**Lemma 1** (Informal). *For $t$ fixed, the asymptotic dynamics of $Z_A^t$ and $B_t$ follow the recursive formula*

$$\begin{aligned} \gamma[Z_A^t] &= \max(\gamma[Z_A^{t-1}], \gamma[\eta] + 1) \\ \gamma[B_t] &= \max(\gamma[B_{t-1]}], \gamma[\eta]). \end{aligned} \tag{3}$$

The formal proof of Lemma 1 is provided in Appendix A and relies on Assumption 1 which fairly represents practical scenarios (see Appendix A for a detailed discussion). Lemma 1 captures the change in asymptotic behavior of quantities $Z_A^t$ and $B_t$ as width grows. Naturally, the dynamics depend on the the initialization scheme which lead to completely different behaviors as we show in the next two results.

### 3.5 Init[A] leads to more efficient feature learning but suffers "internal" instability

Next, we provide a precise characterization of stability and feature learning when using Init[A].

**Theorem 1** (Informal). *For t fixed, with Init[A] and learning rate $\eta$, we have*

---

[8]The $\gamma$-operator is a mapping from the set $\{v, \text{ s.t. } v = \Theta(n^{\beta}) \text{ for } \beta \in \mathbb{R} \cup \{-\infty\}\}$ to the set $\mathbb{R} \cup \{-\infty\}$.

- _**Stability**_: $Z_B^t = \mathcal{O}(1)$ _if and only if_ $\gamma[\eta] \leq -1/2$.

- _**Feature Learning**_: $\Delta Z_B^t = \Theta(1)$ _if and only if_ $\gamma[\eta] = -1/2$. _In this case, we also have_ $\overline{\delta_t^1}, \delta_t^2 = \Theta(1)$ _(efficient feature learning, Definition 5)._

_Moreover, "internal" instability ($Z_A^t = \Omega(1)$) occurs when $\gamma[\eta] \in (-1, 1/2]$._

With `Init[A]`, the maximal learning rate[9] that does not lead to instability in $Z_B$ scales as $\Theta(n^{-1/2})$. This can be seen as an asymptotic form of the edge of stability phenomenon [17] where if we increase the learning rate beyond some level, instability occurs. Interestingly, in this case (i.e. with $\Theta(n^{-1/2})$ learning rate) the features are efficiently updated (Definition 5). However, this comes with caveat: the features $Z_A^t$ grow as $\Theta(n^{1/2})$ which can potentially cause numerical instabilities. We call this phenomenon _internal instability_: only the features $Z_A$ (internal LoRA features) grows, LoRA output $Z_B$ remains $\Theta(1)$ in this case.

The fact that $\Theta(n^{-1/2})$ is the maximal learning rate that does not cause instability in $Z_B$ does not mean it is the _optimal_ learning rate. As the width $n$ grows, this internal instability in $Z_A$ will become more and more problematic. Intuitively, we expect that a trade-off appears in this case: the optimal learning rate (found by grid search) to be larger than $\Theta(n^{-1})$ but smaller than $\Theta(n^{-1/2})$, i.e. the network will try to achieve a balance between optimal feature learning ($\gamma[\eta] = -1/2$) and internal stability $Z_A^t = \Theta(1)$ ($\gamma[\eta] = -1$). We verify this empirically in the next section.

### 3.6 `Init[B]` **leads to suboptimal feature learning with internal stability**

In the next result, we show that the maximal learning rate allowed with `Init[B]` is different from that with `Init[A]`, leading to completely different dynamics.

**Theorem 2** (Informal). _For $t$ fixed, with_ `Init[B]`, _we have_

- _**Stability**_: $Z_B^t = \mathcal{O}(1)$ _if and only if_ $\gamma[\eta] \leq -1$.

- _**Feature Learning**_: $\Delta Z_B^t = \Theta(1)$ _if and only if_ $\gamma[\eta] = -1$.

_Moreover, efficient feature learning cannot be achieved with_ `Init[B]` _for any choice of learning rate scaling_ $\gamma[\eta]$ _(that does not violate the stability condition). More precisely, with_ $\Theta(n^{-1})$ _learning rate, the limiting dynamics (when $n \to \infty$) are the same if $B$ was not trained and $A$ is trained._

With `Init[B]`, the maximal learning rate (that does not violate stability) scales as $\Theta(n^{-1})$ (for any $\epsilon > 0$, a learning rate of $\Theta(n^{-1+\epsilon})$ leads to $Z_B = \Omega(1)$).

Because of this bound on the maximal learning rate, no internal instability occurs with `Init[B]`. In this case, feature learning is suboptimal since the $B$ weight matrix is undertrained in the large width limit ($\delta_t^2 \to 0$).

**Conclusions from Sections 3.5 and 3.6.** The results of Theorem 1 and Theorem 2 suggest that `Init[A]` allows the use of _larger learning rates_ compared to `Init[B]`, which might lead to better feature learning and hence better performance at the expense of some internal instability. Here, 'larger' learning rate should be interpreted in asymptotic terms: with `Init[A]` the maximal learning rate that does not cause instability satisfies $\gamma[\eta] = -1/2$. With `Init[B]`, we have $\gamma[\eta] = -1$ instead. Note that because of the constants in $\Theta(n^\beta)$ learning rates (for some $\beta$), the optimal learning rate with `Init[A]` is not systematically larger than `Init[B]` for _finite width_. However, as width grows, we will see that it is case.

Another important insight from this analysis is that with both initializations, the dynamics are suboptimal in the limit: internal instability with `Init[A]` and undertraining of $B$ with `Init[B]`.[10] We will later discuss possible solutions to this behavior.

---

[9]Maximal $\gamma[\eta]$ that does not cause instability in $Z_B$

[10]More precisely, one can show that with `Init[B]`, for fixed $t$, in the limit $n \to \infty$, $B_t$ converges to $B_0$, i.e. $B$ is untrained in this limit.

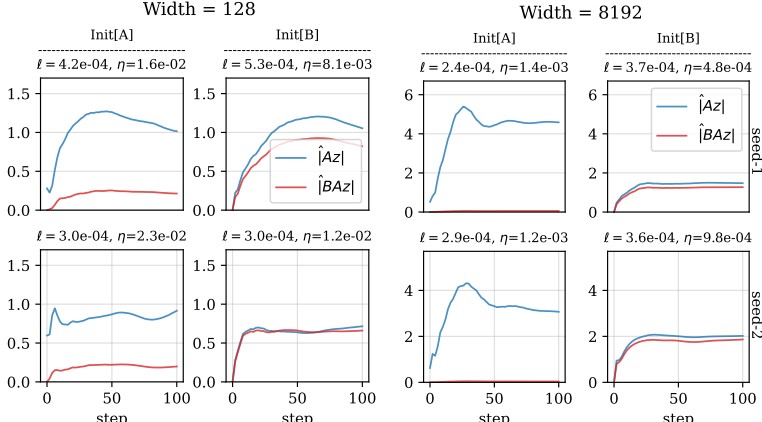

Figure 3: Evolution of the norms of the $Z_A$, $Z_B$ features, averaged over training data. We compute the average $\hat{|Z_A|} \overset{def}{=} N^{-1} \sum_{i=1}^{N} \|Z_A(x_i)\|$ (and same for $Z_B$), where the $x_i$'s are the training data. The dynamics are shown for widths $n = 128$ and $n = 8192$, two seeds, and for both `Init[A]` and `Init[B]`. Train loss and the (optimal) learning rate are shown on top of each plot. We observe that the magnitude of $Z_A$ is significantly higher with `Init[A]` compared to `Init[B]` at large width ($n = 8192$). Interestingly, the train loss is smaller with `Init[A]`, as compared to `Init[B]`. Results with other seeds and widths are shown in Appendix B.

## 3.7 Toy Model

To validate our theory in a controlled setting, we consider the following simple model:

$$\begin{cases} Y_{in} = W_{in}x, \\ Y_h = Y_{in} + (W_h + BA)\phi(Y_{in}) \quad (4) \\ Y_{out} = W_{out}\phi(Y_h) \end{cases}$$

where $W_{in} \in \mathbb{R}^{n \times d}, W_h \in \mathbb{R}^{n \times n}, W_{out} \in \mathbb{R}^{1 \times n}$, and $B, A^\top \in \mathbb{R}^{r \times n}$.

We generate synthetic data from the teacher model using the following config: $d = 5, r_{teacher} = 20, n = 1000, N = 1000$ (train data size), and $N_{test} = 100$ (test data size). The weight $W_{in}^{teacher}, W_{out}^{teacher}, A^{teacher}$, and $B^{teacher}$ are randomly initialized, and $W_h^{teacher} = 0$.[11] We train student models with $d = 5, r = 4$, and varying widths $n \in \{2^k, k = 7, \dots, 13\}$.[12]

**Optimal Learning Rate.** We finetune model (4) on synthetic data generated from the teacher model. In Figure 2, we show the optimal learning rate when using either `Init[A]` or `Init[B]` to initialize the fine-

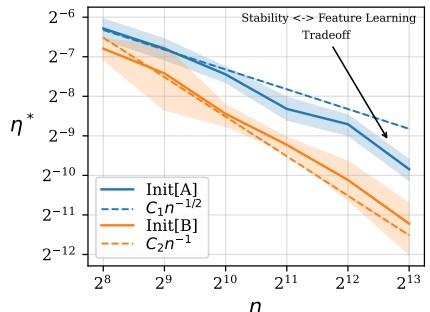

Figure 2: Optimal Learning rate for the fine-tuning of synthetic model Equation (4) with `Init[A]` and `Init[B]` as initialization. The optimal LRs are shown as a function of width $n$. Theoretical lines $n^{-1}$ and $n^{-1/2}$ are shown as well (constants $C_1, C_2$ are chosen to provide suitable trend visualization). As model width $n$ grows, the optimal learning rate with `Init[A]` becomes larger than the optimal learning rate with `Init[B]`. This is in agreement with the theoretical results.

tuning, as a function of width $n$. For $n \gg 1$ (typically $n \geq 2^9$), the optimal learning rate with `Init[A]` is larger than the optimal learning rate with `Init[B]`. This is in agreement with the theoretical results obtained in Theorem 1 and Theorem 2 which predict asymptotic maximal learning rates (that satisfy the stability condition) of $\Theta(n^{-1/2})$ and $\Theta(n^{-1})$ respectively.

With `Init[A]`, we observe the stability/feature learning trade-off for large $n$. The optimal learning rate with `Init[A]` in this regime (e.g. $n = 2^{13}$) is smaller than the maximal theoretical learning

---

[11]Here, the pretrained model is effectively given by $Y_{out} = W_{out}^{teacher}\phi(W_{in}^{teacher}x)$, and the finetuning dataset is simulated by injecting the LoRA weights $A^{teacher}, B^{teacher}$.

[12]In this setup, a student model can have larger width $n$ than the teacher model.

rate $n^{-1/2}$ that achieves optimal feature learning (Theorem 1). Here, the model seems to balance the internal instability that occurs in the $Z_A$ features with feature learning and thus favors smaller learning rates: the optimal learning rates is smaller than $\Theta(n^{-1/2})$ and larger than $\Theta(n^{-1})$.

**Internal Instability and Feature Learning.** Figure 3 shows the average magnitude of $Z_A$ and $Z_B$ for `Init[A]` and `Init[B]` for widths $n \in \{128, 8192\}$. With `Init[A]`, the magnitude of $Z_A$ features seem to grow with width, hence trading off internal stability for more efficient feature learning. This behavior is consistent across random seeds as shown in the figure, and as further confirmed by experiments in Appendix B. The train loss is consistently smaller with `Init[A]`, which can be explained by the fact that `Init[A]` allows more efficient feature learning at the cost of some internal instability. This flexibility cannot be achieved with `Init[B]`. Note also that $Z_B$ features tends to get smaller with $n$ with `Init[A]` as predicted by theory: the trade-off between internal instability and feature learning implies that $\eta^* = o(n^{-1/2})$, which implies that $Z_B^t = o(1)$, i.e. the $Z_B$ features vanish as width grows. While this might problematic, it only becomes an issue when the width is extremely large: if the optimal learning rate scales as $\Theta(n^{-\beta})$ for some $\beta \in (1/2, 1)$ (so that the learning rate is between $\Theta(n^{-1})$ and $\Theta(n^{-1/2})$, balancing internal instability and efficient feature learning), the LoRA output feature scales as $Z_B = B_t A_t \underline{Z} = \Theta(n^{-\beta+1})$. Therefore, if $\beta \approx 0.7$ for instance, the vanishing rate of LoRA output feature is $Z_B \approx \Theta(n^{-0.3})$ which is slow given the order of magnitude of width in practice (for $n = 2^{12}$, we have $n^{-0.3} \approx 0.08$).

## 4 Experiments with Language Models

Our theoretical results from earlier provides a detailed asymptotic analysis of the finetuning dynamics when LoRA modules are initialized with `Init[A]` or `Init[B]`. The main conclusions are that `Init[A]` generally leads to more efficient feature learning (which can be justified by the fact that optimal learning rate is larger when using `Init[A]` compared to when using `Init[B]`). To provide evidence of this claim on real-world tasks, we use LoRA to finetune a set of language models on different benchmarks. Details about the experimental setup and more empirical results are provided in Appendix B. We use LoRA+ code [44] for our experiments (available at https://github.com/nikhil-ghosh-berkeley/loraplus).

### 4.1 GLUE tasks with RoBERTa

The GLUE benchmark (General Language Understanding Evaluation) consists of several language tasks that evaluate the understanding capabilities of langu8age models [8]. Using LoRA, we finetune Roberta-large from the RoBERTa family [12] on MNLI, SST2, and QNLI tasks with varying learning rates $\eta$ and initialization schemes (`Init[A]` or `Init[B]`). We use the same experimental setup of [19] for Roberta-Large to compare our results with theirs (see Appendix B for more details).

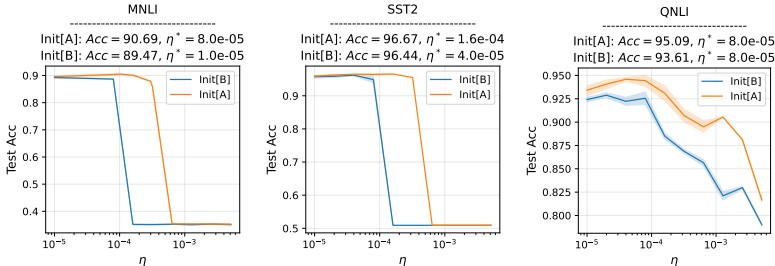

Figure 4: Test Accuracy for RoBERTa-Large finetuned on GLUE tasks. The results are shown after convergence of finetuning with LoRA, initialized with either `Init[A]` or `Init[B]`. Models were finetuned using LoRA rank $r = 8$ and FP16 precision. Optimal learning rate and corresponding accuracy are shown on top of each panel for both initializations. The experimental setup is provided in Appendix B.

The results in Figure 4 are aligned with our theory: we observe that `Init[A]` generally leads to better performance, and the optimal learning rate with `Init[A]` is generally larger than with `Init[B]`. Models initialized with `Init[A]` match the performances reported in [19], while those initialized with `Init[B]` generally underperform that baseline. For MNLI task (the hardest one amongst the

three tasks), we observe a significant difference in the best test accuracy (over 3 random seeds) with $90.69$ with `Init[A]` and $89.47$ with `Init[B]`. We also observe that for MNLI, the optimal learning rate with `Init[A]` ($\eta^* = 8$e-5) is much larger than the optimal learning rate with `Init[B]` ($\eta^* = 1$e-5), which aligns with our theoretical predictions. However, note that for QNLI for instance (an easier task), while the optimal test accuracy is significantly better with `Init[A]`, the optimal learning rate (from the grid search) is the same for `Init[A]` and `Init[B]`. There are many possible explanations for this: 1) the width is not large enough in this case to see the gap between optimal learning rates (for RoBERTa-Large, the width is $n = 2^{10}$) 2) The constants in $\Theta(n^{-1})$ are $\Theta(n^{-1/2})$ are significantly different in magnitude due to dependence on finetuning task. We notice similar behaviour with LLama experiments below. A precise analysis of this observation is beyond the scope of this paper, we leave it for future work.

## 4.2 Llama

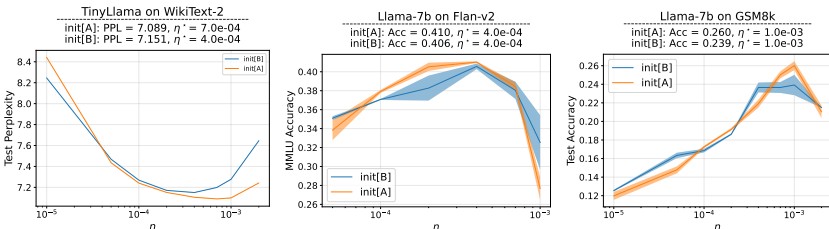

Figure 5: **(Left)** Test perplexity (lower is better) of TinyLlama LoRA on WikiText-2 with `Init[A]` and `Init[B]`. **(Center)** MMLU accuracy of Llama-7b LoRA finetuned on the Flan-v2 dataset. **(Right)** GSM8k test accuracy of Llama-7b LoRA finetuned on the GSM8k dataset. More experimental details are provided in Appendix B.

To further validate our theoretical findings on more modern models and datasets, we report the results of finetuning the Llama-7b model [38] on the Flan-v2 dataset [36] and the GSM8k dataset [16], and finetuning the TinyLlama model [49] on WikiText-2 using LoRA. Each trial is averaged over two seeds and the shaded region indicates one standard error. In the left panel of Figure 5 we see that when finetuning TinyLlama using LoRA the optimal learning rate using `Init[A]` is larger than with `Init[B]` and the corresponding test perplexity is lower. Similarly, for the center panel of Figure 5, when finetuning the Llama-7b model on Flan-v2, the optimal learning rates for `Init[A]` and `Init[B]` are the same (for the learning rate grid we used), but the the optimal MMLU accuracy for `Init[A]` is slightly higher than for `Init[B]`. For learning rates close to the optimal choice, the accuracy using `Init[A]` is generally higher than for `Init[B]`. An analagous result holds for the GSM8k dataset as shown in the rightmost panel of Figure 5. More details about this setting are provided in Appendix B.

## 5 Conclusion and Limitations

We showed that LoRA dynamics are highly sensitive to initialization. `Init[A]` is associated with larger optimal learning rates, compared to `Init[B]`. Larger learning rates typically result in better performance, as confirmed by our empirical results. Note that this is a zero-cost adjustment with LoRA finetuning: *we simply recommend using `Init[A]` instead of `Init[B]`.*

One limitation of our work is that we only define feature learning via the magnitude of feature updates in the limit of large width. In this way, our definition of feature learning is data-agnostic and therefore no conclusion about generalization can be obtained with this analysis. The constants in $\Theta(.)$ asymptotic notation naturally depend on the data (the finetuning task) and therefore such data-agnostic approach does not allow us to infer any information about the impact of the data on the finetuning dynamics.

*More importantly*, our results indicate that both initialization schemes lead to suboptimal scenarios, although `Init[A]` allows more efficient feature learning. In both cases, instability and/or suboptimal feature learning present fundamental issues, which can potentially be mitigated by approaches such as LoRA+ [44]. Understanding the interaction of LoRA+ and related efficiency methods with the initialization scheme is an important question for future work.

# 6 Acknowledgement

We thank Gradient AI for cloud credits under the Gradient AI fellowship awarded to SH and thank AWS for cloud credits under an Amazon Research Grant awarded to the Yu Group. We also gratefully acknowledge partial support from NSF grants DMS-2209975, 2015341, 20241842, NSF grant 2023505 on Collaborative Research: Foundations of Data Science Institute (FODSI), the NSF and the Simons Foundation for the Collaboration on the Theoretical Foundations of Deep Learning through awards DMS-2031883 and 814639, and NSF grant MC2378 to the Institute for Artificial CyberThreat Intelligence and OperatioN (ACTION).

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

# A Theory and Proofs

## A.1 Role of A and B weight matrices

Recall the feature update decomposition

$$\Delta Z_B^t = \underbrace{B_{t-1}\Delta Z_A^t}_{\delta_t^1} + \underbrace{\Delta B_t Z_A^{t-1}}_{\delta_t^2} + \underbrace{\Delta B_t \Delta Z_A^t}_{\delta_t^3}. \tag{5}$$

To achieve optimal feature learning, we want to ensure that $\delta_t^1 = \Theta(1)$ and $\delta_t^2 = \Theta(1)$ which means that both weight matrices $A$ and $B$ are efficiently updated and contribute to the update in $Z_B$. To justify why this is a desirable property, let us analyze how changes in matrices $A$ and $B$ affect LoRA feature $Z_B = BA\underline{Z}$.

Let $(B_{:,i})_{1 \leq i \leq r}$ denote the columns of $B$. We have the following decomposition of $Z_B$:

$$Z_B = \sum_{i=1}^{r} (A\underline{Z})_i B_{:,i},$$

where $(A\underline{Z})_i$ is the $i^{th}$ coordinate of $A\underline{Z}$. This decomposition suggests that the *direction* of $Z_B$ is a weighted sum of the columns of $B$, and $A$ modulates the *weights*. With this, we can also write

$$\begin{cases} \delta_t^1 = \sum_{i=1}^r (\Delta A_t \underline{Z})_i (B_{:,i})_{t-1} \\ \delta_t^2 = \sum_{i=1}^r (A_{t-1}\underline{Z})_i (\Delta B_{:,i})_{t-1}, \end{cases}$$

where $(B_{:,i})_t$ refers to the columns of $B$ at time step $t$. Having both $\delta_t^1$ and $\delta_t^2$ of order $\Theta(1)$ means that both $A$ and $B$ are 'sufficiently' updated to induce a change in weights $(A\underline{Z})_i$ and directions $B_{:,i}$. If one of the matrices $A, B$ is not efficiently updated, we might end up with suboptimal finetuning, leading to either non updated directions $B$ or direction weights $(A_{t-1}Z)$. For instance, assuming that the model is initialized with Init[B], and that $B$ is not efficiently updated, the direction of $Z_B$ will be mostly determined by the vector (sub)space of dimension $r$ generated by the columns of $B$ at initialization.

This intuition was discussed in details in [44].

## A.2 Scaling of Neural Networks

Scaling refers to the process of increasing the size of one of the ingredients in the model to improve performance (see e.g. [22]). This includes model capacity which can be increased via width (embedding dimension) or depth (number of layers) or both, compute (training data), number of training steps etc. In this paper, we are interested in scaling model capacity via the width $n$. This is motivated by the fact that most state-of-the-art language and vision models have large width.

It is well known that as the width $n$ grows, the network initialization scheme and the learning should be adapted to avoid numerical instabilities and ensure efficient learning. For instance, the initialization variance should scale $1/n$ to prevent arbitrarily large pre-activations as we increase model width $n$ (e.g. He init [4]). To derive such scaling rules, a principled approach consist of analyzing statistical properties of key quantities in the model (e.g. pre-activations) as $n$ grows and then adjust the initialization, the learning rate, and the architecture itself to achieve desirable properties in the limit $n \to \infty$ [5, 10, 13].

In this context, Yang et al. [26] introduces the Maximal Update Parameterization (or $\mu$P), a set of scaling rules for the initialization scheme, the learning rate, and the network architecture that ensure stability and maximal feature learning in the infinite width limit. Stability is defined by $Y_l^i = \Theta(1)$ for all $l$ and $i$ where the asymptotic notation '$\Theta(.)$' is with respect to width $n$ (see next paragraph for a formal definition), and feature learning is defined by $\Delta Y_l = \Theta(1)$, where $\Delta$ refers to the feature update after taking a gradient step. $\mu$P guarantees that these two conditions are satisfied at any training step $t$. Roughly speaking, $\mu$P specifies that hidden weights should be initialized with $\Theta(n^{-1/2})$ random weights, and weight updates should be of order $\Theta(n^{-1})$. Input weights should be initialized $\Theta(1)$ and the weights update should be $\Theta(1)$ as well. While the output weights should be

initialized $\Theta(n^{-1})$ and updated with $\Theta(n^{-1})$. These rules ensure both stability and feature learning in the infinite-width limit, in contrast to standard parameterization (exploding features if the learning rate is well tuned), and kernel parameterizations (e.g. Neural Tangent Kernel parameterization where $\Delta Y_l = \Theta(n^{-1/2})$, i.e. no feature learning in the limit).

### A.3 When does $\gamma$-Operator fail to capture asymptotic behavior?

When non-polynomial dependencies (in terms of $n$) appear in neural computations, then the $\gamma$ operator cannot capture asymptotic behavior of the learning dynamics. For instance, if one of the layers has embedding dimension $e^n$ or $n \times \log(n)$, polynomial exponents are no longer sufficient to capture the asymptotic dynamics. Fortunately, such cases are generally not considered in practice.

### A.4 Proof of Lemma 1

In this section, we provide the formal proof of Lemma 1. The proof relies on the following assumption on the processed gradient $g_A$. This assumption was used in [44] to derive scaling rules for the optimal learning rates for $A$ and $B$ weight matrices. Here, we use it to study the sensitivity of LoRA dynamics to initialization. We provide an intuitive discussion that shows why this assumption is realistic.

**Assumption 1.** *With the same setup of Section 3, at training step $t$, we have $\underline{Z}, d\bar{Z} = \Theta(1)$ and $g_A^t \underline{Z} = \Theta(n)$.*

Assumption 1 consists of two parts: that 1) $\underline{Z}, d\bar{Z} = \Theta(1)$ and 2) $g_A^t \underline{Z} = \Theta(n)$. The first condition is mainly related to pretraining paramterization which we assume satisfied such conditions.[13] The second condition is less intuitive, so let us provide an argument to justify why it is sound in practice. Let us study the product $g_A^t \underline{Z}$ in the simple case of Adam with no momentum, a.k.a SignSGD which is given by

$$g_A = \text{sign}\left(\frac{\partial \mathcal{L}}{\partial A}\right),$$

where the sign function is applied element-wise. At training step $t$, we have

$$\frac{\partial \mathcal{L}_t}{\partial A} = \frac{\alpha}{r} B_{t-1}^\top d\bar{Z}^{t-1} \otimes \underline{Z},$$

Let $S^t = \frac{\alpha}{r} B_{t-1}^\top d\bar{Z}^{t-1}$. Therefore we have

$$g_A = \text{sign}(S^t \otimes \underline{Z}) = (\text{sign}(S_i^t \underline{Z}_j))_{1 \leq i,j \leq n}.$$

However, note that we also have

$$\text{sign}(S_i^t \underline{Z}_j) = \text{sign}(S_i^t)\text{sign}(\underline{Z}_j),$$

and as a result

$$g_A^t = \text{sign}(S^t) \otimes \text{sign}(\underline{Z}).$$

Hence, we obtain

$$g_A^t \underline{Z} = (\text{sign}(\underline{Z})^\top \underline{Z})\text{sign}(S^t) = \Theta(n),$$

where we used the fact that $\text{sign}(\underline{Z})^\top \underline{Z} = \Theta(n)$.

This intuition should in-principle hold for the general variant of Adam with momentum as long as the gradient processing function (a notion introduced in [2]) roughly preserves the $\text{sign}(\underline{Z})$ direction. This reasoning can be made rigorous for general gradient processing function using the Tensor Program framework and taking the infinite-width limit where the components of $g_A, \underline{Z}, d\bar{Z}$ all become

---

[13]There is a technical intricacy on this point. While $\underline{Z}$ depends only on pretraining, the Jacobian $d\bar{Z}$ depends on finetuning. However, under the stability conditions mentioned in Definition 3, if $d\bar{Z} = \Theta(1)$, it should remain so during finetuning as well.

iid. However this necessitates an intricate treatment of several quantities in the process, which we believe is an unnecessary complication and does not serve the main purpose of this paper.

**Lemma 1.** *Under Assumption 1, the asymptotic behaviour of $Z_A^t$ and $B_t$ follow the recursive formula*

$$\gamma[Z_A^t] = \max(\gamma[Z_A^{t-1}], \gamma[\eta] + 1)$$
$$\gamma[B_t] = \max(\gamma[B_{t-1]}], \gamma[\eta]).$$

*Proof.* At finetuning step $t$, the weights are updated as follows

$$A_t = A_{t-1} - \eta g_A^{t-1}, \quad B_t = B_{t-1} - \eta g_B^{t-1}.$$

Using the elementary operations with the $\gamma$-operator, we obtain

$$\gamma[Z_A^t] = \max(\gamma[Z_A^{t-1}], \gamma[\eta g_A^{t-1} \underline{Z}]) = \max(\gamma[Z_A^{t-1}], \gamma[\eta] + \gamma[g_A^{t-1} \underline{Z}]).$$

We conclude for $Z_A^t$ using Assumption 1. The formula for $\gamma[B_t]$ follows using the same techniques.

□

## A.5   Proof of Theorem 1

**Theorem 1.** *Under Assumption 1, For $t$ fixed, with* `Init[A]` *and learning rate $\eta$, we have*

- *Stability: $Z_B^t = \mathcal{O}(1)$ if and only if $\gamma[\eta] \leq -1/2$.*

- *Feature Learning: $\Delta Z_B^t = \Theta(1)$ if and only if $\gamma[\eta] = -1/2$. In this case, we also have $\overline{\delta_t^1}, \delta_t^2 = \Theta(1)$ (efficient feature learning, Definition 5).*

*Moreover, "internal" instability ($Z_A^t = \Omega(1)$) occurs when $\gamma[\eta] \in (-1, 1/2]$.*

*Proof.* With `Init[A]`, we have $\gamma[B_0] = -\infty$ and $\gamma[A_0 \underline{Z}] = 0$. As a result, we have for all $t$

$$\gamma[A_t \underline{Z}] = \max(0, \gamma[\eta] + 1)$$
$$\gamma[B_t] = \gamma[\eta]$$

To achieve $Z_B = \mathcal{O}(1)$, we should therefore have

$$\gamma[\eta] + \max(0, \gamma[\eta] + 1) \leq 0,$$

which is equivalent to $\gamma[\eta] \leq -1/2$.

This implies that the maximum learning rate that does not cause instability is $\Theta(n^{-1/2})$. Such learning rate causes internal instability, i.e. the feature $Z_A$ explodes with width. Why? Because, with this learning rate, we have $\gamma[A_t \underline{Z}] = 1/2$, i.e. $A_t \underline{Z} = \Theta(n^{1/2})$ which diverges as $n$ grows. However, this growth is compensated with the fact that $\gamma[B_t] = -1/2$, i.e. $B_t = \Theta(n^{-1/2})$. This analysis is valid for any $\gamma[\eta] \in (-1, 1/2]$.

In this case, feature learning is efficient in the sense of Definition 5: $\delta_t^1 = \Theta(1)$ and $\delta_t^2 = \Theta(1)$. To see this, recall that $\delta_t^1 = B_{t-1} \Delta Z_A^1$ which yields $\gamma[\delta_t^1] = \gamma[B_{t-1}] + \gamma[\Delta Z_A^t] = \gamma[\eta] + \gamma[\eta] + 1 = 0$ and $\gamma[\delta_t^2] = \gamma[\Delta B_t] + \gamma[Z_A^{t-1}] = \gamma[\eta] + \max(\gamma[\eta] + 1, 0) = 0$. So both weights contribute significantly to feature updates at the expense of benign exploding in $Z_A^t = A_t \underline{Z}$.

□

### A.6 Proof of Theorem 2

**Theorem 2.** *Under Assumption 1, for $t$ fixed, with `Init[B]` and learning rate $\eta$, we have*

- *Stability: $Z_B^t = \mathcal{O}(1)$ if and only if $\gamma[\eta] \leq -1$.*
- *Feature Learning: $\Delta Z_B^t = \Theta(1)$ if and only if $\gamma[\eta] = -1$.*

*Moreover, efficient feature learning cannot be achieved with `Init[B]` for any choice of learning rate scaling $\gamma[\eta]$ (that does not violate the stability condition). More precisely, with $\Theta(n^{-1})$ learning rate, the limiting dynamics (when $n \to \infty$) are the same if $B$ was not trained and $A$ is trained.*

*Proof.* Here, we show that maximal learning rate that does not cause instability in LoRA output features $Z_B$ is $\Theta(n^{-1})$ and no internal instability occurs in this scenario.

With `Init[B]`, we have that $\gamma[B_0] = 0$ and $\gamma[A_0\underline{Z}] = -\infty$. From Equation (3), we obtain that

$$\gamma[A_t\underline{Z}] = \gamma[\eta] + 1$$
$$\gamma[B_t] = \max(0, \gamma[\eta]).$$

As a result, LoRA output stability is achieved if and only if

$$\gamma[\eta] + 1 + \max(0, \gamma[\eta]) \leq 0,$$

which is equivalent to having $\gamma[\eta] \leq -1$.

Moreover, with $\eta = \Theta(n^{-1})$ we have that $\gamma[\delta_t^1] = \gamma[B_{t-1}] + \gamma[\Delta Z_A^t] = 0 + \gamma[\eta] + 1 = 0$ and $\gamma[\delta_t^2] = \gamma[\Delta B_t] + \gamma[Z_A^{t-1}] = \gamma[\eta] + 0 = -1$. As a result, feature learning is not efficient in this case, and the learning dynamics are asymptotically equivalent to not training matrix $B$ (because $\delta_t^2 \to 0$). $\qquad\square$

## B   Additional Experiments

This section complements the empirical results reported in the main text. We provide the details of our experimental setup, and show the acc/loss heatmaps for several configurations.

### B.1   Empirical Details

#### B.1.1   Toy Example

In Figure 2, we trained a simple model with LoRA layers to verify the results of the analysis in **??**. Here we provide the empirical details for these experiments.

**Model.**   We consider a simple model given by

$$f(x) = W_{out}\phi(W_{in}x + (W_h + BA)\phi(W_{in}x)),$$

where $W_{in} \in \mathbb{R}^{n \times d}, W_{out} \in \mathbb{R}^{1 \times n}, A \in \mathbb{R}^{r \times n}, B \in \mathbb{R}^{n \times r}$ are the weights, and $\phi$ is the ReLU activation function.

**Dataset.**   Here, we used $d = 5$, $n = 1000$, and $r = 20$ to simulate synthetic data (the teacher model). Synthetic dataset generated by $X \sim \mathcal{N}(0, I_d), Y = f(X)$. The number of training examples is $N_{train} = 1000$, and the number of test examples is $N_{test} = 100$. the weights $W_{in}, W_h, W_{out}, B, A$ are randomly sampled from a Gaussian distribution with normalized variance (1/fan-in).

**Training.**   We train the model with AdamW with $\beta_1 = 0.9$ and $\beta_2 = 0.99$ for a range for values of $\eta$. The weights are initialized as follows: $W_{in} \sim \mathcal{N}(0, 1/d), W_h \sim \mathcal{N}(0, 1/n), W_{out} \sim \mathcal{N}(0, 1/n)$ and fixed. Only the weight matrices $A, B$ are trainable.

### B.1.2 GLUE tasks with RoBERTa

For our experiments with RoBERTa models, finetuned on GLUE tasks, we use the following setup:

**Training Alg Details**

| Model | Roberta-Large |
|---|---|
| **Learning Rates** | $\{2^k \times 10^{-5}, \text{ for } k = 0, 1, 2, \ldots, 10\}$ |
| $\beta_1$ | 0.9 |
| $\beta_2$ | 0.999 |
| $\varepsilon$ | $1 \times 10^{-8}$ |
| **LR Schedule** | Linear with Warmup Ratio 0.06 |
| **Weight Decay** | 0.0 |
| **Train Batch Size** | 4 |
| **Number of Epochs** | 10 |

**LoRA Hyperparameters**

| **LoRA Rank** | 8 |
|---|---|
| **LoRA** $\alpha$ | 16 |
| **LoRA Dropout** | 0.1 |
| **Target Modules** | 'query, value' |

**Other Hyperparameters**

| **Sequence Length** | $T_{\text{target}} = 128$ |
|---|---|
| **Random Seeds** | 3 |
| **Precision** | FP16 |

**GPUs.** Nvidia A10 with 24GB VRAM.

### B.1.3 TinyLlama WikiText-2

For our experiments using the TinyLlama model finetuned on Wikitext-2, we use the following setup training with AdamW.

**Training Algorithm Details**

| | |
|---|---|
| **Learning Rates** | $1 \times 10^{-5}$, $5 \times 10^{-5}$, $1 \times 10^{-4}$, $2 \times 10^{-4}$, $4 \times 10^{-4}$, $7 \times 10^{-4}$, $1 \times 10^{-3}$, $2 \times 10^{-3}$ |
| $\beta_1$ | 0.9 |
| $\beta_2$ | 0.999 |
| $\varepsilon$ | $1 \times 10^{-6}$ |
| **LR Schedule** | Linear with Warmup Ratio 0.03 |
| **Weight Decay** | 0.0 |
| **Train Batch Size** | 8 |
| **Number of Epochs** | 1 |

**LoRA Hyperparameters**

| | |
|---|---|
| **LoRA Rank** | 64 |
| **LoRA** $\alpha$ | 16 |
| **LoRA Dropout** | 0.0 |
| **Target Modules** | 'q_proj, k_proj, v_proj, o_proj, up_proj, down_proj, gate_proj' |

**Other Hyperparameters**

| | |
|---|---|
| **Sequence Length** | 1024 |
| **Random Seeds** | 2 |
| **Precision** | BF16 |

**GPUs.** Nvidia A10 with 24GB VRAM.

### B.1.4 Llama-7b Flan-v2

For our experiments using the Llama-7b model finetuned on a size 100k random subset of flan-v2, we use following setup training with AdamW

**Training Algorithm Details**

| | |
|---|---|
| **Learning Rates** | $1 \times 10^{-5}$, $5 \times 10^{-5}$, $1 \times 10^{-4}$, $2 \times 10^{-4}$, $4 \times 10^{-4}$, $7 \times 10^{-4}$, $1 \times 10^{-3}$ |
| $\beta_1$ | 0.9 |
| $\beta_2$ | 0.999 |
| $\varepsilon$ | $1 \times 10^{-6}$ |
| **LR Schedule** | Linear with Warmup Ratio 0.03 |
| **Weight Decay** | 0.0 |
| **Train Batch Size** | 16 |
| **Number of Epochs** | 1 |

**LoRA Hyperparameters**

| | |
|---|---|
| **LoRA Rank** | 64 |
| **LoRA** $\alpha$ | 16 |
| **LoRA Dropout** | 0.0 |
| **Target Modules** | 'q_proj, k_proj, v_proj, o_proj, up_proj, down_proj, gate_proj' |

**Other Hyperparameters**

| | |
|---|---|
| **Sequence Length** | $T_{\text{source}} = 1536$, $T_{\text{target}} = 512$ |
| **Random Seeds** | 2 |
| **Precision** | BF16 |

**MMLU Evaluation:** We evaluate average accuracy on MMLU using 5-shot prompting.

**GPUs:** Nvidia A10 with 24GB VRAM.

### B.1.5 Llama-7b GSM8k

For our experiments using the Llama-7b model finetuned on the GSM8k training dataset, we use following setup training with AdamW

**Training Algorithm Details**

| | |
|---|---|
| **Learning Rates** | $1 \times 10^{-5}$, $5 \times 10^{-5}$, $1 \times 10^{-4}$, $2 \times 10^{-4}$, $4 \times 10^{-4}$, $7 \times 10^{-4}$, $1 \times 10^{-3}$ |
| $\beta_1$ | 0.9 |
| $\beta_2$ | 0.999 |
| $\varepsilon$ | $1 \times 10^{-6}$ |
| **LR Schedule** | Linear with Warmup Ratio 0.03 |
| **Weight Decay** | 0.0 |
| **Train Batch Size** | 16 |
| **Number of Epochs** | 1 |

**LoRA Hyperparameters**

| | |
|---|---|
| **LoRA Rank** | 64 |
| **LoRA** $\alpha$ | 16 |
| **LoRA Dropout** | 0.0 |
| **Target Modules** | 'q_proj, k_proj, v_proj, o_proj, up_proj, down_proj, gate_proj' |

**Other Hyperparameters**

| | |
|---|---|
| **Sequence Length** | $T_{\text{source}} = 1536$, $T_{\text{target}} = 512$ |
| **Random Seeds** | 2 |
| **Precision** | BF16 |

**GPUs:** Nvidia A10 with 24GB VRAM.

### B.2  Additional Exps

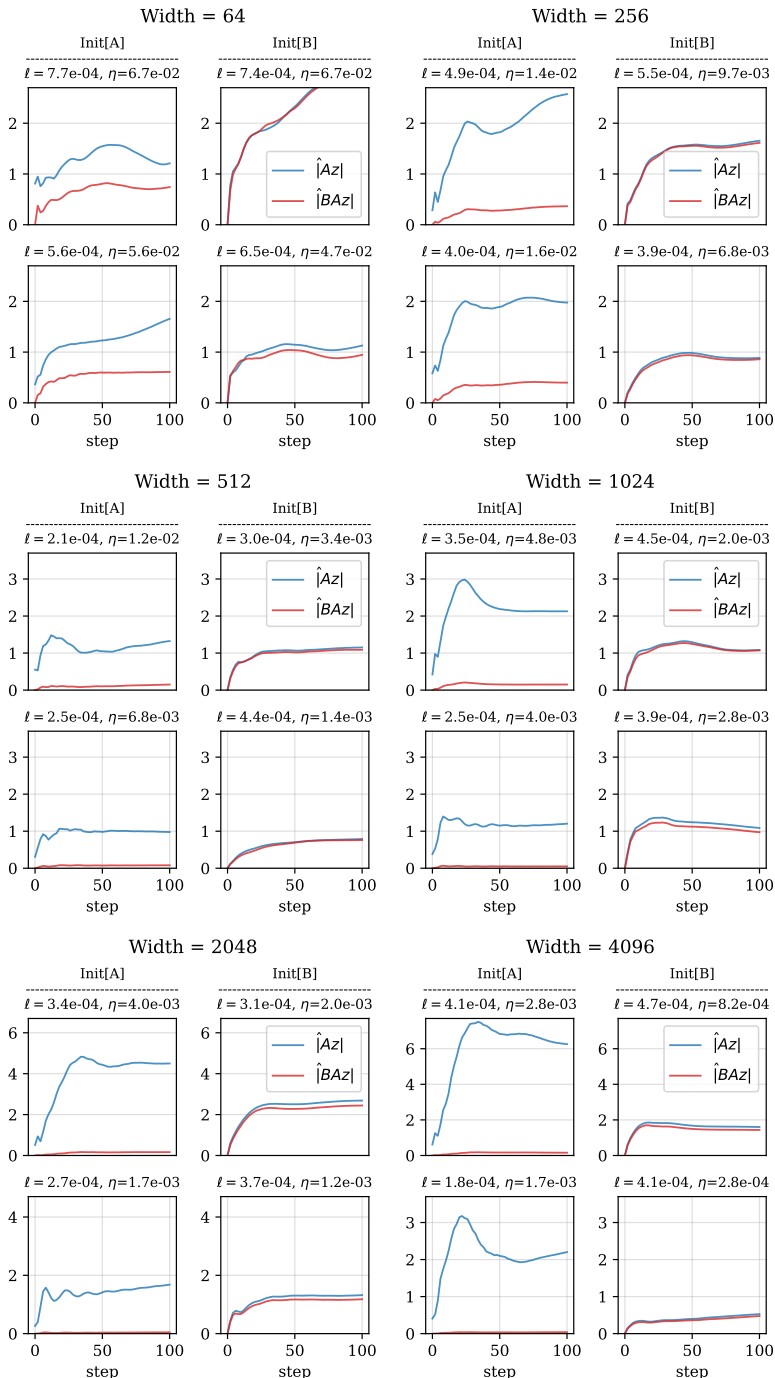

Figure 6: Same as Figure 3 with differents widths.

