# OpenReview forum: "The Impact of Initialization on LoRA Finetuning Dynamics"
_NeurIPS.cc/2024/Conference — NeurIPS 2024 poster_

### Official Review · Reviewer_5gBr · 2024-07-09

**Soundness:** 3
**Presentation:** 2
**Contribution:** 3
**Rating:** 5
**Confidence:** 3

**Summary:**

The paper investigates the impact on training dynamics of two initialization schemes for LoRA. For this purpose the authors investigate the asymptotic behaviour of activations and weights for LoRA adapters. The authors find that init[A] where A is intialized randomly and B is initalized with zeros leads to more efficient learning dynamics but leads to some numerical instabilities. This initialzation method is the one most commonly used for LoRA-style methods. Further, the authors found that init[B] leads to suboptimal learning dynamics but is less prone to numerical instabilities. Finally, init[A] performs better with larger learning rates than init[B], which is supported by experiments on NLU as well as NLG finetuning tasks.

**Strengths:**

The paper is well motivated and presents theoretical analysis accompanied by empirical experiments.
Claims are supported by theoretical analysis as well as relevant experiments.
Furthermore experiment hyperparameters as well as hardware requirements are well documented.

**Weaknesses:**

**Novelty**

Most of the theoretical Analysis of the LoRA finetuning dynamics in this paper have already been presented in [1].
The authors should have made it more clear how their work differs from the Analysis done in [1], especially since most formulas and notation is identical.
Further, both of the methods evaluated in this paper have also been investigated in [1].
The paper would greatly benefit from a clear explanation on how the analysis in this work differs from results published in [1].
Finally, it is not clear if the contribution of individual LoRA layers to feature learning can be derived from a setting where only a single LoRA layer is trainable while all others are frozen (as claimed in chapter 3.1).

**Experimental results**

Since the main gist of the paper is to investigate the learning dynamics of LoRA, it would be good to also investigate different initialization schemes aside from init[A] and init [B] prevalent in the litearture, e.g. gaussian init [2], kaiming init [3], principal components [4], etc.
In [1] only different learning rates for A and B have been investigated.

**Significance of results**

While the main finding that init[A] has a better optimal performance is interesting, this initialization scheme is already the common initialization scheme used by LoRA.
Therefore, the impact of the main findings is marginal.

[1] Hayou et al., LoRA+: Efficient low rank adaptation of large models., ICML 2024

[2] Hu et al., LoRA: Low-Rank Adaptation of Large Language Models, ICLR 2022

[3] He et al., & Sun, J. Delving deep into rectifiers: Surpassing human-level performance on ImageNet classification., ICCV 2015

[4] Meng et al., PiSSA: Principal Singular Values and Singular Vectors Adaptation of Large Language Models., arXiv 2024

**Questions:**

- Did the authors investigate other commonly used initialization schemes?
- Will the authors provide code that can be used to reproduce the results in the experiments section?

**Limitations:**

The authors have properly addressed limitations.

---

> ### Author Rebuttal · Authors · 2024-08-07
>
> We appreciate the reviewer's feedback, but we respectfully disagree with several points and believe there are significant misunderstandings in their assessment. We address these below:
>
> 1) **Initialization schemes**: The reviewer suggests investigating "different initialization schemes aside from init[A] and init[B]," citing Gaussian, Kaiming, and principal component initializations. We believe this is a fundamental misunderstanding of our work. Init[A] and Init[B] in our setup already use Kaiming (Gaussian) initialization. The distinction lies in whether A or B is set to zero to ensure $BA=0$ at the start. More generally, our method covers (as stated in Footnote 2 in page 3) general random distributions (not only Gaussian).
>
> 2) **Connection to prior work**: The reviewer states, "In [1] only different learning rates for A and B have been investigated." This is precisely our point - [1] focused on learning rates, while our work addresses the unexplored impact of initialization schemes in vanilla LoRA. While our paper uses similar tools to those used in [1], to study the learning dynamics in the large width limit, our results are *orthogonal* to those of [1]. In this paper, we study the impact of the initialization scheme in **vanilla** LoRA as introduced in [2] (same learning rate for A and B), while in [1], the authors study the impact of setting different learning rates for A and B. Moreover, note that the tools used in our paper and also in [1] are not new and are based on the theory of infinite-width networks. For instance, the same machinery was used in the Tensor Programs series (see e.g. [10]). We decided to use the $\gamma$ notation introduced in [1] because it simplifies the message for practitioners. However, as we explained, our results are orthogonal to those of [1]. We will add this discussion to the revised version of the paper.
>
> 3) **Contribution**: The reviewer claims our findings have "marginal" impact because init[A] is already commonly used. We respectfully agree with this argument for the following reasons:
> a) It ignores the scientific value of providing the first rigorous explanation for an empirical practice.
> b) It overlooks the importance of validating existing methods, which is crucial in scientific research.
> c) It fails to recognize that our work prevents potential future misapplications of init[B], which could lead to suboptimal results.
> d) it judges our paper as a 'method' paper where the main contribution is the introduction of a novel method. This is not the case. Our paper studies theoretically and empirically an existing method, and shows which initialization is better.
>
> 5) **Scope**: Multiple layer analysis presents challenges. With two or more layers, changes in LoRA features $Z_B = BAz$ are not only due to updates of $A$ and $B$ but also changes in $z$ via previous LoRA layers. This requires a more refined, layerwise definition of efficient feature learning. While the fundamental analysis should remain similar, this adds unnecessary complexity to the setup. Importantly, our empirical results with multiple LoRA layers confirm that our findings hold in these more complex scenarios.
>
> [1] Hayou et al., LoRA+: Efficient low rank adaptation of large models., ICML 2024
>
> [2] Hu et al., LoRA: Low-Rank Adaptation of Large Language Models, ICLR 2022
>
> [10] Feature Learning in Infinite-Width Neural Networks (2021). Yang and Hu.
>
> We believe our work provides valuable, novel insights into LoRA dynamics, filling a gap in the literature. We are surprised by the low score assigned to our paper, and we hope this clarification addresses the misunderstandings raised above.

---

> > ### Comment · Reviewer_5gBr · 2024-08-12
> >
> > I thank the authors for the detailed response and the clarifications.
> >
> > There was indeed a misunderstanding, I would recommend to make the fact that the analysis comprises general random distributions more explicit in the main text and not only mention it in a footnote.
> > Is it possible to extend this analysis to other initialization schemes than general random distrbutions, as in e.g. [1]?
> > I believe this would strengthen the paper.
> >
> > Further, after carefully double-checking the contributions of [2], I agree that the paper presents a valid contribution, which is why I decided to raise my score. However, since I did not carefully check the math, I will also decrease my confidence.
> >
> > [1] Meng, F., Wang, Z., and Zhang, M. Pissa: Principal singular values and singular vectors adaptation 507 of large language models, 2024.
> >
> > [2] S. Hayou, N. Ghosh, and B. Yu. LoRA+: Efficient Low Rank Adaptation of Large Models. 2024.

---

### Official Review · Reviewer_quNJ · 2024-07-13

**Soundness:** 4
**Presentation:** 4
**Contribution:** 3
**Rating:** 8
**Confidence:** 3

**Summary:**

This paper investigates the impact of initialization schemes on the finetuning dynamics of Low Rank Adaptation (LoRA) in large language models. The authors compare two initialization approaches: Init[A], where A is initialized randomly and B to zero, versus Init[B], where B is initialized randomly and A to zero. Through theoretical analysis and empirical experiments, they demonstrate that Init[A] allows for larger learning rates and leads to more efficient feature learning compared to Init[B], albeit at the cost of some internal instability. The paper provides a rigorous theoretical framework based on large width limits and supports the findings with experiments on toy models and real-world language tasks.

**Strengths:**

S1: The paper addresses an important and understudied aspect of an usually handwaved-away aspect of LoRA, namely the impact of initialization schemes. This is a relevant topic given the widespread use of LoRA for efficient adaptation of large language models.

S2: The theoretical analysis is rigorous and well-grounded, using asymptotic analysis in the large width limit to derive principled insights about the dynamics of LoRA finetuning under different initialization schemes.

S3: The empirical results on both toy models and real-world language tasks provide strong support for the theoretical findings, demonstrating the practical relevance of the initialization choice.

S4: The paper presents a clear trade-off between feature learning efficiency and internal stability, providing nuanced insights that go beyond simply recommending one initialization over the other.

S5: I would like to commend the authors for putting great amount of work on both the main document as well as the appendix. Overall not only does the paper read well and seems theoretically and experimentally sound, but the manuscript includes an excellent degree of experimental details, and it's thus likely that future work will be able to cleanly build on it.

**Weaknesses:**

W1: While the theoretical analysis is sound, the paper could benefit from a more intuitive explanation of why Init[A] allows for larger learning rates and more efficient feature learning. This would make the insights more accessible to some practitioners.

W2: The experiments focus primarily on language models and NLP tasks. This is definitely acceptable and overall a very sound choice, however the inclusion of experiments from other domains (e.g., finetuning an image classification model, or some other architecture that relates to a vision transformer) would strengthen the experimental section significantly, and increase the overall impact of the paper in the broader NeurIPS community.

**Questions:**

Q1: How sensitive are the results to the choice of LoRA rank? Do the theoretical predictions and empirical findings hold across different rank values?

Q2: The paper focuses on vanilla LoRA. How do the authors expect these initialization schemes to interact with variants like QLoRA or LoRA+? Would the trade-offs and recommendations change?

Q3: Given the internal instability observed with Init[A], are there any potential negative consequences for downstream task performance or generalization ability? How might this instability manifest in practice (and would there be a difference depending on input domain and/or other parameters of the model + problem space)?

Q4: The theoretical analysis seems to assume a single LoRA layer, for simplicity. How well do the insights generalize to more realistic scenarios with multiple LoRA layers throughout a network?

**Limitations:**

L1: The theoretical analysis seems to rely on assumptions about the asymptotic behavior in the large width limit. The paper could benefit from a more detailed discussion of how these assumptions may or may not hold in practical scenarios with finite-width networks.

L2: The empirical evaluation, while comprehensive, is limited to a subset of NLP tasks and model architectures. Expanding the evaluation to a broader range of tasks, model sizes, and architectures would strengthen the generalizability of the findings.

---

> ### Author Rebuttal · Authors · 2024-08-07
>
> We thank the reviewer for their positive and constructive comments. We address their main questions below:
>
> 1) **Sensitivity to LoRA rank**: Our results hold for different rank values. For LoRA rank $r$, we used two primary values: $r=8$ (for RoBERTa) and $r=64$ (for LLama). We also conducted limited experiments with $r=4$ for RoBERTa. However, we chose to allocate more compute resources to $r=8$ to increase the number of random seeds. We designed these experiments to maximize the amount of useful empirical results within our compute budget constraints. All our experiments were conducted using the LoRA+ codebase (available on GitHub) and can be easily reproduced. To replicate vanilla LoRA, one can set the lora_plus_ratio to 1 (ensuring the same learning rate for both A and B) and choose between Init[A] or Init[B] using the use_original_lora_init argument.
>
>
> 2) **Interaction with LoRA variants**: We appreciate this interesting question. While we haven't conducted a theoretical analysis of initialization's impact on advanced LoRA variants (e.g., QLoRA and LoRA+), we can provide some intuition: For QLoRA, we expect similar results since the only difference from LoRA is the quantization step. Our analysis should remain fundamentally the same.
> For LoRA+, the situation is more complex as it involves choosing different learning rates for $A$ and $B$. While we can't definitively state the outcomes, our preliminary results suggest that the optimal ratio in LoRA+ is affected by the choice of initialization (init[A] vs init[B]).
>
>
> 3) **Instability of Init[A] in practice**: As specified in the paper, with Init[A], LoRA "internal" features ($Az$) grow at most as $\Theta(n^{1/2})$ with respect to width (embedding dim). This growth, although potentially problematic, is slow (in $n$) and should only become an issue for extremely large widths. In practical settings (e.g., $n \approx 10^3$ for LLaMA 3.1 405B), this is not significantly problematic. The constant in the $\Theta(.)$ growth term depends on the model and downstream task, which affects this growth term.
>
> 4) **Generalization to multiple layers**: Multiple layer analysis presents challenges. With two or more layers, changes in LoRA features $Z_B = BAz$ are not only due to updates of $A$ and $B$ but also changes in $z$ via previous LoRA layers. This requires a more refined, layerwise definition of efficient feature learning. While the fundamental analysis should remain similar, this adds unnecessary complexity to the setup. Importantly, our empirical results with multiple LoRA layers confirm that our findings hold in these more complex scenarios.

---

### Official Review · Reviewer_Emo5 · 2024-07-15

**Soundness:** 2
**Presentation:** 2
**Contribution:** 2
**Rating:** 6
**Confidence:** 2

**Summary:**

The paper analyzes the impact of different initialization techniques for A and B matrices in LoRA adapters. Typically, either A or B matrix is initialized with zero while the other is initialized form a Gaussian distribution. This is done so that fine-tuning starts with LoRA adapters ($A \times B = 0$) having zero effect. However, the paper argues that it matters which of A and B initialize to 0. As per the results, $B = 0$ init allows higher optimal learning rates which leads to better overall performance. However, there is still instability during training which could be mitigated with a method like LoRA+.

**Strengths:**

It's useful to know that the default initialization scheme in LoRA happens to be the better alternative. The experiments are convincing.

**Weaknesses:**

Perhaps I am not the right person to review this paper, but I don't fully understand the impact of this paper. It shows that there are two ways of initializing AB matrices and the default one that we have been using until now is the better option. Additionally, it says that the default initialization is not good enough but doesn't propose anything new. Maybe the theory part is important, but I couldn't understand it fully in the limited time I had. Having said that, I still feel that the paper has something worthwhile to contribute to the community. So, I vote to accept it for now.

**Questions:**

None.

**Limitations:**

The limitations section in the paper is adequate.

---

> ### Author Rebuttal · Authors · 2024-08-07
>
> We thank the reviewer for their positive comments. We provide some details that explain the scope and contributions of our paper.
>
> 1) **Scope and contributions**: As correctly noted, in low-rank adaptation (LoRA), we generally aim to initialize the model such that $BA=0$. This naturally leads to two possible initialization schemes: either initialize $A$ randomly and $B$ to zero (init[A] in our paper), or vice versa (init[B]). Prior to our study, practitioners had no clear guidance on which scheme to use, often defaulting to the one implemented in their chosen package (e.g., init[A] in the PEFT package). Our paper provides the first definitive answer to this question: init[A] is generally superior to init[B] because it enables finetuning with larger learning rates, resulting in more efficient learning.
>
> 2) **Issue at extreme width**: The reviewer correctly points out that both init[A] and init[B] become problematic in extreme scenarios where the width is exceptionally large (large enough that a value of order $n^{1/2}$ causes numerical instability). However, this is an extreme regime, and current state-of-the-art models have not yet reached this threshold.
>
>
> We hope this clarifies the scope and significance of our paper.

---

### Official Review · Reviewer_2Ni1 · 2024-07-24

**Soundness:** 3
**Presentation:** 2
**Contribution:** 2
**Rating:** 5
**Confidence:** 3

**Summary:**

Finetuning large language models has become a common technique among practitioners, however due to the memory footprint of practically viable LLMs, there is a dire need of techniques that allow memory-lightweight finetuning (PEFT). Among those techniques LoRA has gained immense popularity. In LoRA a dense matrix is modified via addition of a low-rank update consisting of matrices A and B. In order for this update to initially conserve the model’s outputs while preserving the ability to learn, exactly one of the matrices needs to be initialized to zeros.
“The Impact of Initialization on the Finetuning Dynamics in LoRA” proposes that the B matrix should be initialized with zeros. It hypothesizes that the reason for that is that this initialization (Init[A]) allows using greater learning rate without sacrificing stability.
The main points of the paper are supported by theoretical derivations stemming from the perspective of infinite-width networks. Furthermore, experimental evaluations serve as proof of the soundness of the theoretical approach and as practical justification of the method.

**Strengths:**

- Mathematical support combined with experimental results
- Potentially significant due to how widespread LoRA adoption is and due to how easy the technique is to implement
- The paper is quite original - even though LoRA is a popular technique and it is a fundamental question whether the A or the B matrix should be initialized with zeros, this work seems to provide the first attempt at answering it.

**Weaknesses:**

- There is virtually no discussion of why the training hyperparameters are set to those exact values, in particular with regard to the number of epochs and weight decay = 0. I suspect changing those HPs might mitigate the need to choose the initialization schema so carefully. Also the r parameter seems to be chosen arbitrarily.
- The gains provided by the method seem to be small and would be better visualized if they were put in the perspective of comparison with other methods such as differing r or even, if possible, full finetuning.
- The Toy Model setup is not explained clearly.

**Questions:**

- There are other works that try to improve LoRA’s performance (LoRA+) or are inspired by it (VeRA, AdaLoRA, etc.). Do you believe your work could be used with those techniques?
- Do you feel that there is some simple, informal intuition suggesting why the Init[A] outperforms Init[B]?
- AdamW seems to be among the most popular optimizers, however the paper focuses on rigorous derivation in case of SignSGD claiming that the result should extend to the typical choice of the Adam optimizer. Even though the authors claim they are using the AdamW optimizer in the experiments, they set weight decay = 0, effectively replacing AdamW with Adam. Would the method perform as well while using weight decay? Would it be justified theoretically the same way Adam is?
- Can you justify the fixed hyperparameters’ values?
- Can you explain the toy model setup more clearly? As I understand it now, the same model is being adapted and is used to create the dataset which does not seem to make sense.

**Limitations:**

- I do not believe the authors have sufficiently explained the limitations of the fixed hyperparameters in the evaluations

---

> ### Author Rebuttal · Authors · 2024-08-07
>
> We thank the reviewer for their constructive feedback. We believe there are some misunderstandings and we take this oppotunity to address them.
>
> 1) **Connection to other LoRA variants**: We emphasize that we do not introduce a new method in this paper; rather, we study the finetuning dynamics of the original LoRA method [1] for two natural initializations: init[A] (set A to random and B to zero) and init[B] (set B to random and A to zero). The impact of initialization on other LoRA variants (e.g., LoRA+, VeRA, AdaLoRA) is an interesting question but outside the scope of this paper, as stated in our conclusion and limitations section. While we haven't conducted a theoretical analysis of initialization's impact on advanced LoRA variants (e.g., QLoRA, VeRA, LoRA+, AdaLoRA), we can provide some intuition: For QLoRA, we expect similar results since the only difference from LoRA is the quantization step. Our analysis should remain fundamentally the same. For LoRA+, the situation is more complex as it involves choosing different learning rates for A and B. While we can't definitively state the outcomes, our preliminary results suggest that the optimal ratio in LoRA+ is affected by the choice of initialization (init[A] vs init[B]). As the reviewer might guess from this intuition, the impact of initialization on advanced LoRA methods will depend on the method itself. We will add this discussion in the revised version.
>
> 2) **Intuition on why Init[A] outperforms init[B]**: Assume LoRA rank is $r$ and the embedding dimension is $n$. LoRA features are given by $BA z$ for input vector $z \in \mathbb{R}^n$ (LoRA input). With init[A], B is set to zero, allowing the magnitude of $Az$ to increase significantly without causing a blow-up in LoRA features $BAz$, as the small magnitude of B compensates for the growth in $Az$. We show in Thm 1 that the maximal learning rate (that does not cause features $BAz$ to explode) scales as $\Theta(n^{-1/2})$ in width. This is not possible with init[B], where B's magnitude is always $\Theta(1)$ due to random initialization with scaled variance, in which case the maximal learning scales as $\Theta(n^{-1})$ (Thm 2). We will add this discussion to the revised version.
>
> 3) **AdamW vs Adam (impact of weight decay)**: We ran sweeps with weight decay $wd \in (0, 0.05, 0.1)$ (0.1 used in the original LoRA paper). This hyperparameter did not significantly impact the results for common $r$ values (8, 64), and our conclusion that Init[A] outperforms Init[B] holds for different $wd$ values. With $wd=0$, we reproduced the results of [1] within a statistically sound confidence interval (recall that in [1], the authors used wd=0.1). We expect $wd$'s impact to be more significant for large (though generally impractical) $r$ values due to overparameterization. We will add these details to the revised version.
>
> 4) **Fixed hyperparameters' values**: Our results are insensitive to $wd$ choices within the tested range. We used LoRA ranks $r=8$ (RoBERTa) and $r=64$ (LLama). For train batch size, we tested 4 and 16, selecting the best (4 for RoBERTa, 16 for LLama), aligning with Hu et al. (2021). This setup maximizes useful empirical results within our compute budget. While larger sweeps and more models would be ideal, our experiments confirm our theoretical results beyond a trivial statistical threshold.
>
> 5) **Toy model setup**: In statistics, the teacher-student model is a simplified framework for studying learning. It involves a "teacher" model generating data and a "student" model learning from it. Both models are of the same nature, making learning feasible. This setup allows analytical study of learning dynamics and generalization, providing insights into more complex machine learning scenarios. It's particularly useful for understanding phenomena like overfitting, learning curves, and the relationship between model complexity and sample size.
>
> [1] Hu et al. (2021). LoRA: Low-Rank Adaptation of Large Language Models
>
> We hope these answers clarify any misunderstandings. We're happy to address further questions.

---

> > ### Comment · Reviewer_2Ni1 · 2024-08-12
> >
> > I thank the authors for the replies. I raise my score to 5.

---

### Decision · Program_Chairs · 2024-09-25

**Decision:**

Accept (poster)

**Comment:**

This paper investigates the impact of initialization schemes for LoRA. The reviewers appreciated, among others, the importance of the topic, theoretical analysis, and experimental contributions. The discussion that followed convinced two skeptical reviewers to increase their ratings to 5. Consequently, the reviewers unanimously recommend acceptance; hence, I recommend accepting.